Microbiology **Spectrum**

∂ | **Open Peer Review** | Host-Microbial Interactions | Research Article

# Impact of semen microbiota on the composition of seminal plasma

D. Baud,[1,2] A. Zuber,[1] A. Peric,[3] N. Pluchino,[4] N. Vulliemoz,[5] M. Stojanov[1,2]

**ABSTRACT** Several studies have found associations between specific bacterial genera and semen parameters. Bacteria are known to influence the composition of their niche and, consequently, could affect the composition of the seminal plasma. This study integrated microbiota profiling and metabolomics to explore the influence of seminal bacteria on semen metabolite composition in infertile couples, revealing associations between specific bacterial genera and metabolite profiles. Amino acids and acylcarnitines were the predominant metabolite groups identified in seminal plasma. Different microbiota profiles did not result in globally diverse metabolite compositions in seminal plasma. Nevertheless, levels of specific metabolites increased in the presence of a dysbiotic microbiota. Urocanate was significantly increased in abnormal semen samples (adjusted *P*-value < 0.001) and enriched in samples dominated by *Prevotella* spp. (*P*-value < 0.05), which was previously linked to a negative impact on semen. Therefore, varying microbiota profiles can influence the abundance of certain metabolites, potentially having an immunomodulatory effect, as seen with urocanate.

**IMPORTANCE** Male infertility is often considered idiopathic since the specific cause of infertility often remains unidentified. Recently, variations in the seminal microbiota composition have been associated with normal and abnormal semen parameters and may, therefore, influence male infertility. Bacteria are known to alter the metabolite composition of their ecological niches, and thus, seminal bacteria might affect the composition of the seminal fluid, crucial in the fertilization process. Our research indicates that distinct seminal microbiota profiles are not associated with widespread changes in the metabolite composition of the seminal fluid. Instead, the presence of particular metabolites with immunomodulatory functions, such as urocanate, could shed light on the interplay between seminal microbiota and variations in semen parameters.

**KEYWORDS** seminal microbiota, seminal fluid, infertility

More couples are using assisted reproductive technologies to conceive and this can be partly attributed to the observed decrease in sperm quality observed over the past decades (1, 2). While intracytoplasmic sperm injection during *in vitro* fertilization (IVF) is used to circumvent this issue, the procedure remains expensive, invasive, and carries certain risks. It is, therefore, of the utmost importance to further study the causes of male infertility, since in most cases, this condition is considered idiopathic (3). Elements, such as environmental and lifestyle factors, may contribute to poor semen quality (4, 5). Other potential causes remain unexplored, including the effect of the genital microbiota on male (in)fertility.

The human body is vastly colonized by diverse bacteria, which can profoundly influence its physiology (6, 7). While most studies have focused on the female genital tract and its vaginal microbiota, the male genital microbiota has been relatively neglected, with only a limited number of studies conducted, mainly due to the

Address correspondence to M. Stojanov, milos.stojanov@chuv.ch.

The authors declare no conflict of interest.

See the funding table on p. 10.

difficulties in investigating the upper male genital tract. Previous studies on the male genital tract microbiota have primarily focused on males in infertile couples, as semen samples used for semen analysis are readily available (8–11). The presence of members of the *Lactobacillus* genus has often been associated with positive semen parameters, although their exact role remains to be fully elucidated (9, 11–13). On the other hand, several bacterial genera found in semen may interfere with semen quality, as in the case of *Prevotella* spp., which has been negatively associated with semen motility and morphology in multiple studies (9, 11, 14, 15). It is not fully understood how different bacterial species may influence spermatozoa physiology. Early evidence of negative bacterial effects have been shown in the case of sexually transmitted bacteria, including *Mycoplasma genitalium*, *Mycoplasma hominis* (16), *Ureaplasma urealyticum* (17), and *Chlamydia*-related organisms (18). Generally, direct exposure to these pathogens leads to decreased spermatozoa motility (17–19) or increased apoptosis (20, 21).

In addition to direct effects, bacteria can significantly influence their ecological niches through their metabolic activities (22–24). A well-known example is the human vagina, which maintains an acidic pH in a healthy state due to lactobacilli activity (25). Nevertheless, this type of effect of seminal microbiota on semen parameters is still unknown. Seminal fluid mainly consists of secretions from the seminal vesicles and the prostate (26). Its well-established role is to provide a protective and nourishing environment for spermatozoa during their transport to the oocyte for fertilization (26). Multiple studies have assessed metabolite composition of seminal plasma, mainly in patients diagnosed with male infertility (27–31). However, a consensus on the metabolite composition is yet to be reached, due to the heterogeneity of protocols and analysis methods used (32). This is similar for specific associations of metabolites with spermiogram parameters, although some signatures, including enrichment of acylcarnitines in normospermic men, could be observed (27–29, 31).

Differences in metabolite composition of seminal plasma may be influenced by multiple factors, such as genetic background, environmental, and lifestyle factors. Moreover, composition of seminal plasma may be modified by activity of bacteria found in semen. In this study, we integrated the results obtained by two high-throughput methods, microbiota profiling and metabolomics, to gain new understanding of the role of seminal bacteria on sperm physiology and male infertility.

## MATERIALS AND METHODS

### Semen samples

The samples analyzed in this study were obtained from the Andrology and Reproductive Biology Laboratory (LABR) of the Lausanne University Hospital between October 2014 and July 2016 (11). The study was carried out in accordance with the recommendations of the Cantonal Human Research Ethics Commission of Vaud (CER-VD), and the protocol was approved by the CER-VD (protocol 265-14), according to the Swiss Federal Act on Research involving Human Beings. Full information about the research project was given to all patients, and their written consent was obtained to participate in the study. Routine semen assessment was performed according to the WHO guidelines (World Health Organization, 2010). Semen was collected by masturbation after 2 to 5 days of sexual abstinence and examined following 30 min of liquefaction at 37°C. The samples were manually evaluated for volume and pH, and then assessed using optical microscopy for concentration and morphology. Evaluation of concentration and motility (total and progressive) were performed using the computer-assisted sperm analysis tool, CASA SCA (5.4, Microptic SL, Barcelona, Spain). In this study, 20 samples were randomly selected from the initial 94 samples (11).

Aliquots of the samples were filtered through a 0.22-µm polyvinylidene fluoride (Merck Millipore AG, Switzerland) filter using a vacuum-assisted filtration system. The filtered samples were then stored at −80°C until further analysis.

## Metabolite extraction and analysis

Metabolite analysis was performed by the Metabolomics Platform, Faculty of Biology and Medicine, University of Lausanne, Switzerland. Filtered seminal fluid samples (19 µL) were extracted and homogenized by the addition of 75 µL of argon-purged methanol. Homogenized extracts were centrifuged for 15 min at $4,000 \times g$ at 4°C (Hermle, Gosheim, Germany). The resulting supernatant was collected and evaporated to dryness in a vacuum concentrator (LabConco, Missouri, USA). Dried sample extracts were resuspended in methanol:$H_2O$ (4:1, vol/vol) and analyzed by liquid chromatography with tandem mass spectrometry (LC-MS/MS).

## Protein quantification

Evaporation and lysis of protein pellets was performed with 20 mM Tris-HCl (pH 7.5), 4 M guanidine hydrochloride, 150 mM NaCl, 1 mM $Na_2$EDTA, 1 mM EGTA, 1% Triton, 2.5 mM sodium pyrophosphate, 1 mM beta-glycerophosphate, 1 mM $Na_3VO_4$, and 1 µg/mL of leupeptin using the Cryolys Precellys 24 sample Homogenizer ($2 \times 20$ s at 10,000 rpm, Bertin Technologies, Rockville, Maryland, USA) with ceramic beads. The total protein concentration was determined using the BCA Protein Assay Kit (Thermo Scientific, Massachusetts, USA), with absorbance measured at 562 nm using a Hidex spectrophotometer (Hidex, Turku, Finland).

## Multiple pathway-targeted analysis

Extracted samples were analyzed by hydrophilic interaction liquid chromatography coupled to tandem mass spectrometry in both positive and negative ionization modes using a 6495 triple quadrupole system interfaced with a 1290 UHPLC system (Agilent Technologies) (33, 34).

For positive mode analysis, the chromatographic separation was performed using an Acquity BEH Amide column with dimensions of 100 mm × 2.1 mm internal diameter (I.D.) and particle size of 1.7 µm (Waters, Massachusetts, USA). The mobile phase consisted of two components: A, which was composed of 20 mM ammonium formate and 0.1% formic acid (FA) in water, and B, which was 0.1% FA in acetonitrile (ACN). A linear gradient elution was employed, starting with 95% B from 0 to 1.5 min, followed by a decrease to 45% B from 1.5 to 17 min. These conditions were held for 2 min, and then, the initial chromatographic conditions were maintained for an additional 5 min for column re-equilibration. The flow rate was set at 400 µL/min, the column temperature at 25°C, and the sample injection volume at 2 µL. The electrospray ionization (ESI) source conditions were optimized as follows: the dry gas temperature was set to 290°C, the nebulizer pressure at 35 psi, and the flow rate at 14 L/min. The sheath gas temperature was maintained at 350°C with a flow rate of 12 L/min. The nozzle voltage was set to 0 V, and the capillary voltage was set to 2,000 V. The acquisition mode employed was dynamic multiple reaction monitoring (DMRM) with a total cycle time of 600 ms. Optimized collision energies specific to each metabolite were applied during the analysis.

For negative mode analysis, a SeQuant ZIC-pHILIC column with dimensions of 100 mm × 2.1-mm I.D. and particle size of 5 µm (Merck, Darmstadt, Germany) was utilized. The mobile phase consisted of two components: A, which was composed of 20 mM ammonium acetate and 20 mM $NH_4OH$ in water at pH 9.7, and B, which was 100% ACN. A linear gradient elution was employed, starting with 90% B from 0 to 1.5 min, followed by a decrease to 50% B from 8 to 11 min, and further reduction to 45% B from 12 to 15 min. Finally, the initial chromatographic conditions were maintained for a post-run period of 9 min for column re-equilibration (35). The flow rate was set at 300 µL/min, the column temperature at 30°C, and the sample injection volume at 2 µL. ESI source conditions for negative mode analysis were optimized as follows: the dry gas temperature was set to 290°C with a flow rate of 14 L/min, the sheath gas temperature was maintained at 350°C with a flow rate of 12 L/min, and the nebulizer pressure was set to 45 psi. The nozzle

voltage was set to 0 V, and the capillary voltage was set to −2,000 V. The acquisition mode is employed by DMRM with a total cycle time of 600 ms. Optimized collision energies specific to each metabolite were applied during the analysis.

## Quality control

Regular analysis of pooled quality control (QC) samples, which were representative of the entire sample set, was conducted at periodic intervals, typically after every five samples, throughout the entire analytical run. The purpose of this analysis was threefold: first, to assess the quality of the data generated; second, to correct any signal intensity drift that may have occurred during the LC-MS analysis (typically a decrease in signal intensity over time due to sample interaction with the instrument); and third, to identify and remove peaks that exhibited poor reproducibility, as indicated by a coefficient of variation (CV) greater than 30% (35, 36).

Additionally, a series of diluted QC were prepared by diluting the original QC sample with methanol. The dilution levels included 100% QC, 50% QC, 25% QC, 12.5% QC, and 6.25% QC. The selection of metabolites for analysis also considered their linear response across the diluted QC series, ensuring that metabolites exhibiting a consistent and predictable response at different dilution levels were included in the analysis.

## Data processing

The raw LC-MS/MS data were analyzed using the Agilent Quantitative analysis software (version B.07.00, MassHunter Agilent Technologies). The relative quantification of metabolites was performed based on the extracted ion chromatogram areas for the monitored multiple-reaction monitoring transitions.

The resulting tables, which contained the peak areas of the detected metabolites across all samples, were exported to the R software. Within the MRMPROBS software, signal intensity drift correction was applied, and noise filtering was performed if necessary, using a criterion of a CV greater than 30% for QC features (37). This step helped to ensure the reliability of the data by removing peaks with poor reproducibility and reducing any unwanted variability caused by instrument drift or background noise.

## Impact of urocanate on spermatozoa physiology

Washed semen samples were obtained from the LABR. Spermatozoa concentration was adjusted to $10^6$ cells/mL in PureSperm Wash medium (Nidacon, Gothenburg, Sweden). Experiments were carried out in 24-well low-binding plates (Nunclon Sphera, 174930, Thermo Fisher Scientific, Basel, Switzerland). Different concentrations of urocanate (Sigma-Aldrich Chemie, Buchs, Switzerland) or mock were added to the PureSperm Wash medium, and plates were incubated at 37°C with 5% $CO_2$.

The content of each well was transferred in Eppendorf tubes and centrifuged at $500 \times g$ for 7 min. The pellet was then resuspended in 500 µL of Cells Staining Buffer (Biolegend Europe BV, Uithoorn, the Netherlands). After a second round of centrifugation, the pellet was resuspended in 500 µL of Annexin V staining buffer (Biolegend Europe BV). Staining was performed using Zombie Green Fixable Viability Kit (Biolegend Europe BV) and MitoSpy Red (Biolegend Europe BV). After 20 min of incubation at RT in the dark with gentle mixing every 5 min, cells were washed again with 1 mL of Cells Staining Buffer. Samples were fixed in phosphate-buffered saline (PBS) supplemented with 4% paraformaldehyde and incubated for 2 min in the dark. Finally, samples were washed once and resuspended in 100 µL of PBS.

Flow cytometry measurements were performed using a BD Accuri C6 Flow Cytometer (BD Biosciences). Viability (Zombie Green) was recorded with the FL-1 channel, while red fluorescence (MitoSpy Red) was recorded with the FL-3 channel. For each sample, 10,000 events were recorded based on the spermatozoa population. Results were analyzed with the BD Accuri C6 Software (BD Biosciences).

## Data analysis

Data processing and statistical analysis were performed in R and Python. Data visualization was performed with the R package "ggplot2." The detailed analysis pipeline and raw data are available on the following link: https://github.com/dfmemicrobiota/seminal_metabolite. Differential abundance analysis of metabolites was performed using the DESeq2 Bioconductor package (38), and results were visualized with the R package EnhancedVolcano (39). Dimensionality reduction was performed by principal component analysis (PCA) using the FactoMineR R package (40). Heatmap analysis was performed using the Spearman correlation coefficient and the pheatmap R package (41).

## RESULTS

### Description of metabolites in seminal fluid

The metabolite profile was identified in 19 of the initial 20 samples, as one sample failed to meet the quality process criteria. Table 1 depicts the demographic characteristics and andrological parameters of the patients included in the study.

Among the 19 samples, amino acids and analogs constituted the main class of metabolites, accounting for approximately 50% of the total metabolite content (Fig. 1A).

Acylcarnitines were also highly represented, making up more than 25% of the total metabolite content (Fig. 1A). Acetylcarnitine was the most abundant metabolite, comprising 11.6% of total metabolite counts. Among the top 20 metabolites identified in the samples, 12 were classified as amino acids, peptides, and analog subclass (Fig. 1B). Of the top 20 metabolites, five were acylcarnitines. Other metabolites included N-acetylmannosamine, citrate, and carnitine.

### Association of metabolites with spermiogram analysis

We next investigated the differential abundance of specific metabolites or metabolite classes and semen parameters. Through our analysis, we aimed to identify metabolites that were either positively or negatively associated with these parameters. Our results, presented in Fig. 2, revealed that only urocanate was significantly enriched (more than fourfold) in semen samples that had at least one semen parameter under the reference values.

Furthermore, we found that 3-methylhistamine was significantly enriched in samples with lower levels of both progressive and total motility. Interestingly, both urocanate and 3-methylhistamine are involved in the degradation pathways of histidine, an amino acid that we found to be highly abundant in semen (Fig. 1).

Of note, we did not observe any significant difference when metabolites were grouped into classes and subclasses of molecules. The PCA revealed no significant

**TABLE 1** Demographic and clinical characteristics of the patients

| | All samples (n = 19) | Lactobacillus sp.-enriched samples (n = 10) | Prevotella sp.-enriched samples (n = 9) |
|---|---|---|---|
| Age (years) | 34.3 (23.2–53.1) | 34.4 (23.2–44.6) | 34.1 (26.3–53.1) |
| Semen volume (mL) | 3.9 (1.8–6.0) | 3.8 (2.3–6.0) | 3.9 (1.8–5.8) |
| Sperm concentration (mio/mL) | 28.1 (0–120) | 38.7 (1.5–120.0) | 16.6 (0–40.0) |
| Azoospermia (%) | 5.3 | 0 | 11.1 |
| Polyspermia (%) | 15.8 | 20.0 | 11.1 |
| Oligozoospermia (%) | 36.8 | 30.0 | 44.4 |
| Asthenozoospermia (%) | 42.1 | 30.0 | 55.6 |
| Teratozoospermia (%) | 57.9 | 15.8 | 88.9 |
| Infertility length (years) | 1.9 (0.5–5.0) | 1.5 (0.5–2.0) | 2.4 (1.0–5.0) |
| Andrological or urological diseases (%) | 42.1 | 40.0 | 44.4 |
| Smoking (%) | 31.6 | 40.0 | 22.2 |
| Lactobacillus spp. (relative abundance) | 0.231 (0–0.670) | 0.436 (0.195–0.669) | 0.002 (0–0.10) |
| Prevotella spp. (relative abundance) | 0.194 (0–0.532) | 0.001 (0–0.006) | 0.409 (0.283–0.532) |

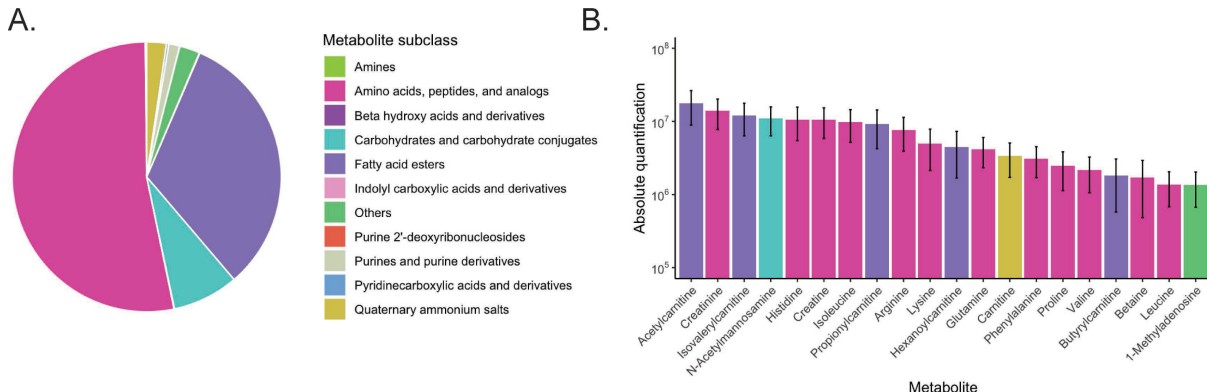

**FIG 1**  Metabolites in the seminal fluid. (A) Pie chart showing the abundance of the main metabolite subclasses. (B) Absolute quantification of the 20 most abundant metabolites identified in seminal fluid. Error bars represent the standard error based on the 19 samples included in the study.

differences in the metabolite composition of seminal plasma when comparing patients with normal and abnormal semen parameters (Fig. S1). This further supports the observation that there were no global variations in the metabolite profiles between these two groups.

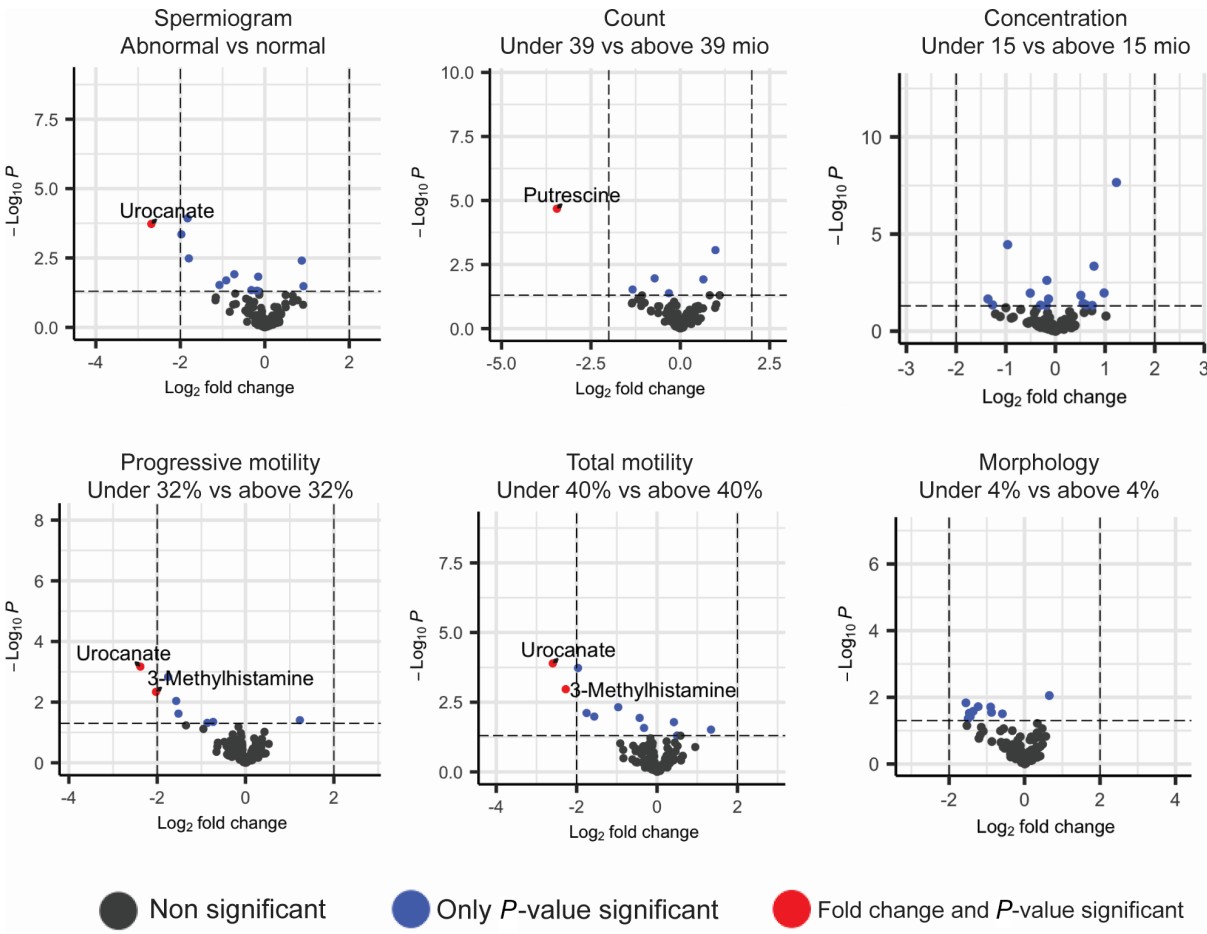

**FIG 2**  Differentially abundant metabolites between patients with spermiogram parameters under and above the reference values set by the WHO. Volcano plot comparison of metabolites in patients with abnormal (left) and normal spermiogram analyses (right) identifies enrichment of specific metabolites via fold change. The plots display the adjusted *P* values (*y*-axis) against the effect sizes (log2 fold change) of each metabolite on the *x*-axis.

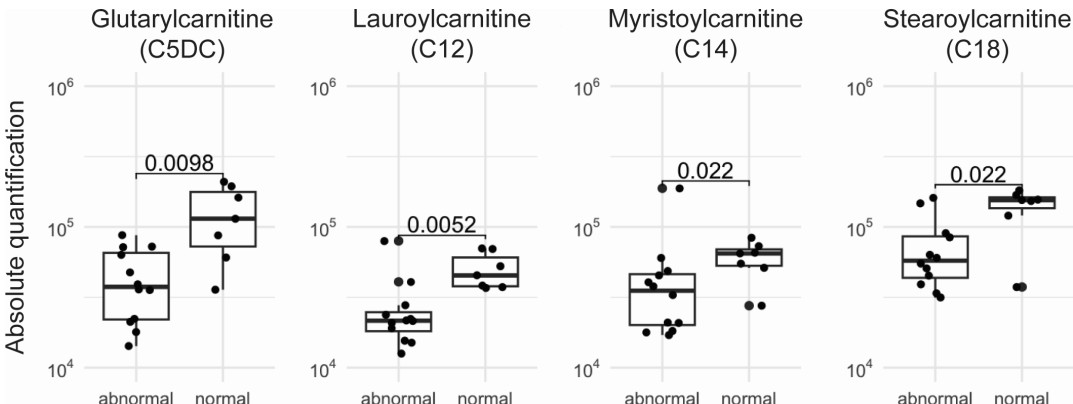

FIG 3 Increased levels of acylcarnitines in semen of patients with normal spermiogram. Comparison of selected acylcarnitines in patients with abnormal and normal spermiogram evaluations. Unpaired Wilcoxon's signed-rank test was employed to evaluate the differences in metabolite abundance between the two groups (*P*-value is shown).

We did not observe any specific enrichment regarding semen analysis for amino acids, the most abundant class of metabolites. Despite not being identified as more than fourfold enriched, levels of several members of acylcarnitines, the second most abundant class of metabolites, were significantly increased in semen from patients with normal semen assessment, compared to those in patients that had at least one parameter under the reference values (Fig. 3).

## Association of metabolites and microbiota data

We have previously determined the bacterial profiles of the semen samples included in this study (11). Since *Prevotella* and *Lactobacillus* were mutually exclusive dominant genera, seminal microbiota profiles could be classified as *Lactobacillus* and *Prevotella* dominant. We initially sought to assess whether there was a global difference in metabolite profiles between the two types of seminal microbiota. Principal component analysis showed that there was no significant difference in the metabolite profile between the two groups (Fig. 4).

We then aimed to identify possible associations between bacterial genera and metabolite profiles identified in this study. Figure 4 depicts a heatmap that correlates relative abundances of bacterial genera and absolute quantification of metabolites. Interestingly, this caused the bacterial genera to cluster according to our previous classifications of the seminal microbiota (11). The *Prevotella*-dominated cluster included members of *Campylobacter*, *Dialister*, *Finegoldia*, *Haemophilus*, and *Peptoniphilus* genera. The *Lactobacillus*-dominated cluster comprised the *Gardnerella*, *Ureaplasma*, *Staphylococcus* genera, and a member of the Plancoccaceae family. A third cluster was made up of *Corynebacterium*, *Veillonella*, and *Delftia* genera. Similar clustering was observed when all the metabolites were included in the heatmap analysis (Fig. S2). A subset of metabolites, including urocanate, orotate, citrulline, niacin, gulonolactone, and 5-hydroxylysine, showed opposite correlations with respect to the *Prevotella*- and *Lactobacillus*-dominated clusters (Fig. 5). Interestingly, urocanate showed a significant positive correlation with *Prevotella* and was significantly enriched in semen samples with abnormal motility parameters.

## Spermatozoa physiology is not directly affected by urocanate

Since urocanate levels were significantly higher in semen of patients with an abnormal semen assessment, we wanted to assess whether this metabolite has a direct or indirect effect on spermatozoa. We therefore exposed motile spermatozoa to urocanate. We used two different metabolite concentrations, one corresponding to the physiological levels found in urine (42) and a 10-fold higher concentration. Spermatozoa viability and

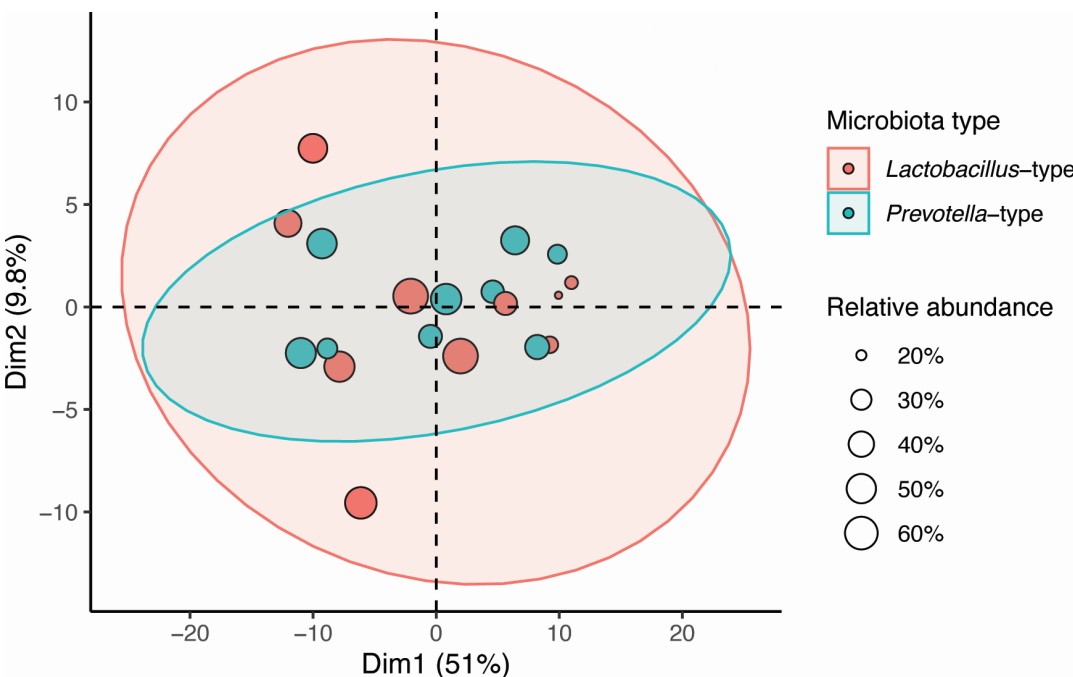

**FIG 4** PCA plot of microbiota samples. The PCA plot visualizes the multidimensional structure of the semen samples based on their metabolite composition. The sample points are colored based on the microbiota type (red, *Lactobacillus* type and blue, *Prevotella* type). The point size represents the abundance of the two genera in the samples. The 95% confidence ellipses depict the distribution of samples within each group.

mitochondrial membrane potential (MMP), which is directly linked to motility (43), were assessed at the beginning of the experiment and after 24 h of incubation. Results were compared to a mock control (Fig. 6).

There were no significant differences between the treatment and control groups after 24 h of incubation for both metabolite concentrations, suggesting that urocanate is not directly impacting spermatozoa physiology.

## DISCUSSION

Research in male infertility has been neglected, despite its significant impact on infertility in approximately half of the affected couples. The cause of male infertility remains unidentified in a vast number of cases, highlighting several gaps in our understanding of this condition (44).

The composition of seminal fluid plays a crucial role in the function of spermatozoa, and several studies have described the presence of microbial communities in the semen (45). With this study, we combined the metabolomic analysis of seminal plasma with the colonization patterns of bacteria in semen samples from infertile couples. Our aim was to evaluate the possible impact of microbial activity on the metabolite composition of semen. Bacteria have the ability to modify their environment through metabolic activities and the production of various compounds, as in the case of multiple human diseases (22). Our results indicate that despite drastically different colonization profiles, there were no major differences in the metabolite composition of the semen samples included in the study.

We observed a high abundance of amino acids in the seminal fluid, which is consistent with previous studies analyzing seminal metabolites (32). Furthermore, several acylcarnitines were enriched in samples exhibiting normal semen parameters, as previously reported (27, 29, 31).

Interestingly, by correlating metabolite quantification with bacterial abundances, we observed major bacterial genera clustering according to profiles previously identified in our seminal microbiota study (11). *Prevotella* spp., highly abundant in semen, clustered

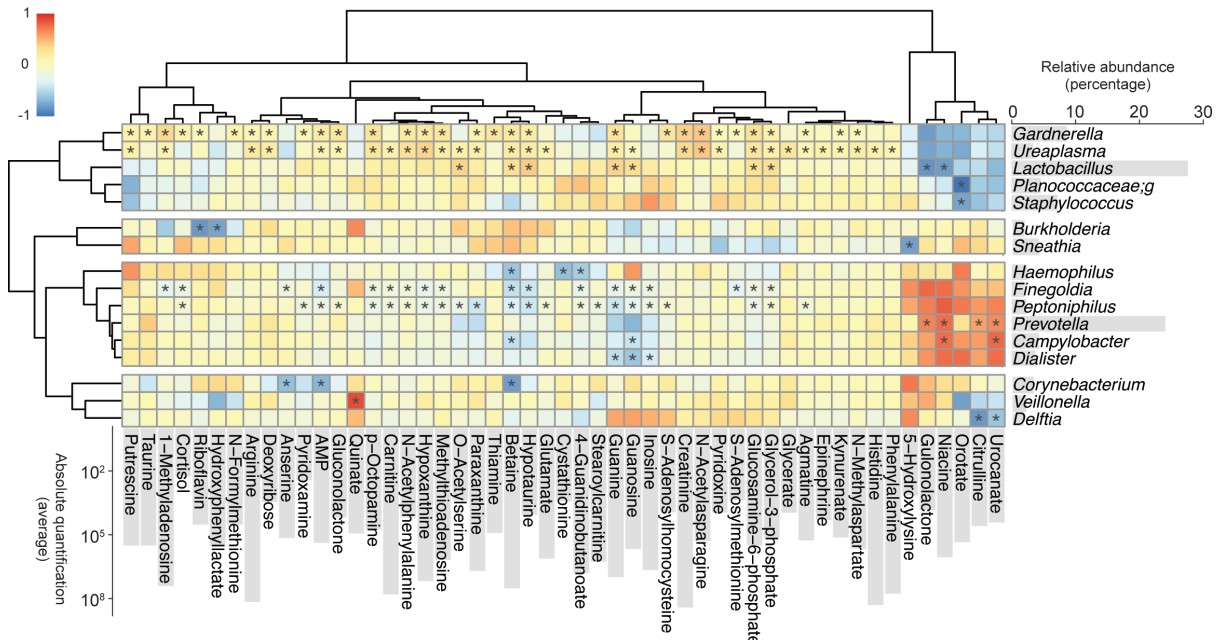

**FIG 5** Correlation of relative abundances of bacterial genera and metabolites in semen. Barplots of the relative abundance of bacterial genera among all samples expressed as percentages is shown on the *y*-axis (left). Barplots of the absolute quantification of metabolites among all samples is shown on the *x*-axis (bottom). Only metabolites that had a correlation coefficient above 0.5 with at least one bacterial genus were retained for the elaboration of the heatmap. Asterisks indicate that correlation is statistically significant ($P < 0.05$) as assessed by the Spearman's rank correlation. The color scale indicates positive (red) or negative (blue) correlations.

with members of the *Campylobacter*, *Dialister*, *Finegoldia*, *Haemophilus*, and *Peptoniphilus* genera. Similarly, *Lactobacillus* spp., also abundant in semen, clustered with *Gardnerella* spp. and *Ureaplasma* spp. This suggests that different bacterial colonization profiles may subtly influence semen metabolites.

We identified several metabolites that were specifically enriched in different seminal microbiota profiles. Notably, urocanate was significantly enriched in a *Prevotella*-dominated profile and in samples with motility values below the reference range. Urocanate has been previously shown to have immunosuppressive and anti-inflammatory functions (46, 47). Interestingly, urocanate is the first intermediate in the degradation pathway

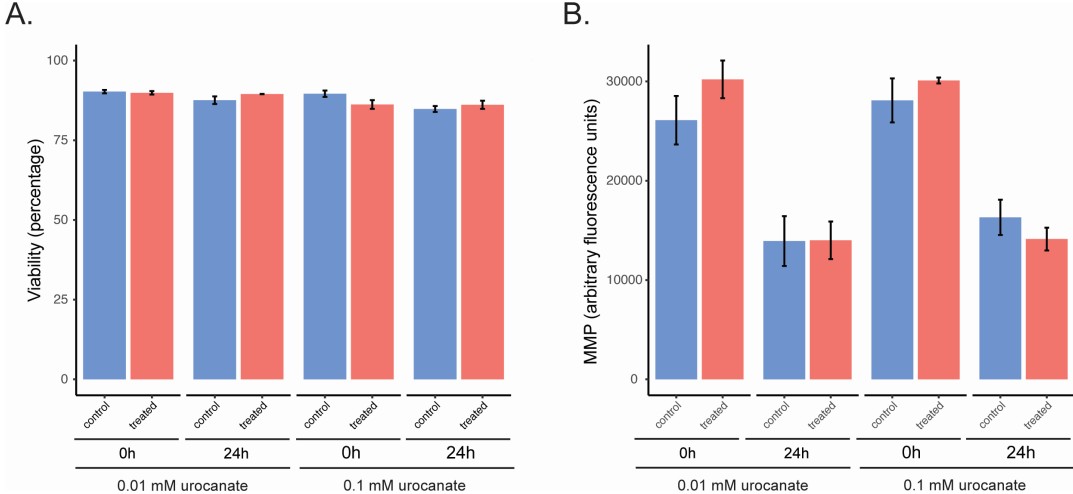

**FIG 6** Effect of urocanate on (A) spermatozoa viability and (B) MMP. Washed spermatozoa were incubated with different concentrations of urocanate and compared to the mock control. Error bars represent the standard deviation of triplicate samples.

of histidine, which was among the most abundant seminal amino acids identified in our study. Additionally, urocanate has been proposed as a potential signaling molecule detected by pathogenic bacteria in the presence of eukaryotic hosts (48).

Several major gaps exist in seminal microbiota research. Longitudinal studies are needed to determine the stability of semen microbiota over time. Furthermore, the exact location of the microbiota in the genital tract remains unknown, a crucial factor for understanding the duration of bacteria's contact with seminal fluid and its potential impact on its composition. Future research should also consider the activity of immune cells together with the metabolite composition of semen and bacterial colonization profiles. Integrating these factors will provide a more comprehensive understanding of the complex interactions between the microbiota, immune system, and metabolites in the context of male infertility.

## ACKNOWLEDGMENTS

We are grateful to all the patients and staff who participated in the study. We thank the Lausanne Genomic Technologies Facility of the University of Lausanne for performing high-throughput sequencing and the Metabolomics Platform of the University of Lausanne for the metabolomic analysis.

This work has been supported by the Woman-Mother-Child Department of the University Hospital of Lausanne.

Conceptualization of the study: D.B. and M.S.; sample collection and processing: D.B., A.Z., and M.S.; data processing and analysis: D.B., A.Z., and M.S.; manuscript draft preparation: D.B., A.Z., A.P., N.V., N.P., and M.S.; manuscript editing: D.B., A.Z., A.P., N.V., N.P., and M.S. All authors have reviewed the manuscript.

## AUTHOR AFFILIATIONS

[1]Materno-Fetal and Obstetrics Research Unit, Mother-Woman-Child Department, University Hospital of Lausanne, Lausanne, Switzerland
[2]Faculty of Biology and Medicine, University of Lausanne, Lausanne, Switzerland
[3]360° Fertility Center Zurich, Zollikon, Switzerland
[4]Fertility Medicine and Gynaecological Endocrinology Unit, Department Woman-Mother-Child, Lausanne University Hospital, Lausanne, Switzerland
[5]CPMA Lausanne, Lausanne, Switzerland

## AUTHOR ORCIDs

M. Stojanov http://orcid.org/0000-0002-5166-5841

## FUNDING

| Funder | Grant(s) | Author(s) |
| --- | --- | --- |
| Schweizerischer Nationalfonds zur Förderung der Wissenschaftlichen Forschung (SNF) | 320030-169853/1 | D. Baud |

## AUTHOR CONTRIBUTIONS

D. Baud, Conceptualization, Funding acquisition, Resources, Supervision, Writing – original draft, Writing – review and editing | A. Zuber, Data curation, Formal analysis, Methodology, Validation, Writing – original draft, Writing – review and editing | A. Peric, Conceptualization, Methodology, Writing – original draft, Writing – review and editing | N. Pluchino, Conceptualization, Writing – original draft, Writing – review and editing | N. Vulliemoz, Conceptualization, Writing – original draft, Writing – review and editing | M. Stojanov, Conceptualization, Data curation, Formal analysis, Investigation, Methodology, Project administration, Resources, Software, Supervision, Validation, Visualization, Writing – original draft, Writing – review and editing

## DATA AVAILABILITY

The detailed analysis pipeline and raw data are available on the following link: https://github.com/dfmemicrobiota/seminal_metabolites.

## ADDITIONAL FILES

The following material is available online.

### Supplemental Material

**Figure S1 (Spectrum02911-23-s0001.docx).** PCA plots of metabolite composition in relation to spermiogram parameters.
**Figure S2 (Spectrum02911-23-s0002.docx).** Correlation of relative abundances of bacterial genera and all the metabolites identified in semen.

### Open Peer Review

**PEER REVIEW HISTORY (review-history.pdf).** An accounting of the reviewer comments and feedback.

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
