## [Reviewer comments · Microbiology Spectrum]

Microbiology Spectrum

Impact of semen microbiota on the composition of seminal plasma

David Baud, Arnaud Zuber, Adriana Peric, Nicola Pluchino, Nicolas Vulliemoz, and Milos Stojanov

Corresponding Author(s): Milos Stojanov, Centre Hospitalier Universitaire Vaudois

Review Timeline:

Submission Date:	July 25, 2023
Editorial Decision:	December 20, 2023
Revision Received:	January 17, 2024
Accepted:	January 20, 2024

Editor: David Pride

Reviewer(s): The reviewers have opted to remain anonymous.

Transaction Report:

DOI: <https://doi.org/10.1128/spectrum.02911-23>

Re: Spectrum02911-23 (Impact of semen microbiota on the composition of seminal plasma)

Dear Dr. Milos Stojanov:

Thank you for the privilege of reviewing your work. Below you will find my comments, instructions from the Spectrum editorial office, and the reviewer comments.

Revision Guidelines

Sincerely,
David Pride
Editor
Microbiology Spectrum

Reviewer #2 (Comments for the Author):

The authors have presented an interesting paper on the microbiome/metabolome of the seminal tract in relation to seminal quality parameters. The work was well done, but is limited by the small sample size here. Some additional comments follow

Major
-Need a table 1, description of the cohort. (how many normal, abnl), demographics, tobacco, obesity... maybe even include a

table of the sperm analysis of all 19 samples

- Missing data on numerous factors already known to impact seminal fluid quality - age, tobacco, EtOH, obesity, etc.. These might be important to control for in any comparisons
- 19 is a small number for a microbiome/metabolomic study
- Print Quality of figures needs to be improved
- Volcano plots are a bit difficult to understand. If the right side of the plot pertains to nl I would highlight the urocanate/putresceine on the nl side as well
- Not clear how there was a strong correlation with Gardnerella and Ureaplasma with multiple metabolites when the fold change was around 0 for many
- Clarify how p-values were obtained (multiple comparison corrections?)

Minor

numerous English corrections needed. Here are just a few

Line 41 missing the word "on" before male fertility

Line 45 "to investigate the" should be "in investigating the"

Line 53 "First" should be "Early"

The authors have presented an interesting paper on the microbiome/metabolome of the seminal tract in relation to seminal quality parameters. The work was well done, but is limited by the small sample size here.

We thank the reviewer for the positive comments. We agree that the limited sample size is a limitation of our study. Nevertheless, we think that our results suggest important conclusions that may be confirmed by future studies involving larger cohorts.

Some additional comments follow:

All the modifications have been highlighted in yellow in the revised version of the manuscript.

Major

-Need a table 1, description of the cohort. (how many normal, abnl), demographics, tobacco, obesity... maybe even include a table of the sperm analysis of all 19 samples

We added a new table with demographic and andrological characteristics of the patients. We agree that this information is relevant to the present study.

- Missing data on numerous factors already known to impact seminal fluid quality - age, tobacco, EtOH, obesity, etc.. These might be important to control for in any comparisons

These analyses were conducted in our previous study, which comprised a larger cohort (PMID: 30809218). Considering the small sample size of the current study, we believe these analyses are not informative here, where we instead focus on the metagenomic and metabolomic aspects of the samples.

- 19 is a small number for a microbiome/metabolomic study

As stated above, we agree with the reviewer on this point. Nevertheless, this is a first report combining both metabolomic and microbiota results and may represent a starting point for future studies about the influence of seminal microbiota on semen physiology.

- Print Quality of figures needs to be improved

We share the same opinion. Unfortunately, the quality of images in the reviewer documents is significantly reduced compared to our original figures. We hope that the images that we uploaded in the new document have an improved quality.

- Volcano plots are a bit difficult to understand. If the right side of the plot pertains to nl I would highlight the urocanate/putresceine on the nl side as well

The volcano plots presented in Figure X are asymmetrical, indicating that each metabolite appears only once in each graph. Consequently, when considering the first volcano plot, all points on the left side represent metabolites that are enriched in samples with abnormal spermiogram, while those on the right side are enriched in samples with normal spermiogram. Metabolites positioned in the exact middle are not significantly enriched in either group. We believe this method of presenting data simplifies the graphs, making them more comprehensible to the readership.

-Not clear how there was a strong correlation with Gardnerella and Ureaplasma with multiple metabolites when the fold change was around 0 for many

Interestingly, these two bacterial genera appear to have a positive and generally weak correlation with the majority of metabolites. Nevertheless, these correlations are significant as assessed by the Spearman's rank correlation coefficient. As an example, we show the correlations between the first four metabolites in the heatmap (Figure 5) and Gardnerella genus:

-Clarify how p-values were obtained (multiple comparison corrections?)

We agree that this should be specified in the manuscript (it was already present in the Github pipeline). This information was added in the legend of Figure 5 (heatmap). Statistical significance of the correlations was calculated by the Spearman's rank correlation coefficient (see previous answer). Briefly, the correlation coefficient is converted into a t-score using a specific formula, comparing this score to a theoretical distribution (the t-distribution), and then using this comparison to determine the probability (p-value) that the observed correlation occurred by chance under the null hypothesis.

Minor

numerous English corrections needed. Here are just a few

Line 41 missing the word "on" before male fertility

Line 45 "to investigate the" should be "in investigating the"

Line 53 "First" should be "Early"

The text has been carefully proofread for English language accuracy.

Re: Spectrum02911-23R1 (Impact of semen microbiota on the composition of seminal plasma)

Dear Dr. Milos Stojanov:

Your manuscript has been accepted, and I am forwarding it to the ASM production staff for publication. Your paper will first be checked to make sure all elements meet the technical requirements. ASM staff will contact you if anything needs to be revised before copyediting and production can begin. Otherwise, you will be notified when your proofs are ready to be viewed.

Sincerely,
David Pride
Editor
Microbiology Spectrum